# Childhood ADHD and Early-Onset Bipolar Disorder Comorbidity: A Case Report

**DOI:** 10.3390/brainsci10110883

**Published:** 2020-11-20

**Authors:** Paraskevi Tatsiopoulou, Georgia-Nektaria Porfyri, Eleni Bonti, Ioannis Diakogiannis

**Affiliations:** 1st Department of Psychiatry, School of Medicine, Faculty of Health Sciences, Aristotle University of Thessaloniki, General Hospital “Papageorgiou”, Ring Road Thessaloniki, N. Efkarpia, 54603 Thessaloniki, Greece; geoporfyri@hotmail.fr (G.-N.P.); elina.bonti@gmail.com (E.B.); idiakogiannis@auth.gr (I.D.)

**Keywords:** early on-set bipolar disorder (BD), attention-deficit/hyperactivity disorder (ADHD), first manic episode, BD-ADHD comorbidity, child and adolescent psychiatry

## Abstract

Introduction: Recent research has highlighted an increased rate of co-morbidity between the neurodevelopmental-behavioral disorder of attention deficit hyperactivity disorder (ADHD) and a variety of psychiatric disorders, such as mood disorders or bipolar disorder (BD). The etiology and clinical course of BD are considered to be determined by both genetic and environmental factors, either aggravating or improving. Aim: This follow-up study of an adolescent aimed to clarify the co-morbidity between ADHD and BD. We also discuss the controversies surrounding the two diagnoses in younger populations and describe several aspects of concern regarding diagnosis, differential diagnosis, therapeutic planning/intervention, and prognosis. Methods: Reporting of a two-year follow-up study of a bipolar 15-year-old female patient with a previous diagnosis of ADHD during childhood. Results: Despite the occurrence of major risk factors, such as early onset and positive family history, the patient’s condition rapidly remitted with medication, without relapse and/or rehospitalization during the following two years, due to the stability of her cooperation, and support of a stable and caring familial environment. Early diagnosis of BD and differential diagnoses of ADHD are considered crucial protective factors leading to an appropriate planning of treatment. In addition, parental involvement and empathic attitude towards the patient supported the latter to cooperate and comply with the treatment, enhancing positive outcomes and stability. Conclusions: Research is required into the reliability and validity of diagnostic protocols and criteria for BD in children and adolescents, and also into the development of individualized therapeutic planning.

## 1. Introduction

Attention deficit hyperactivity disorder (ADHD) is defined in the Diagnostic and Statistical Manual of Mental Disease 5th Edition (DSM-V) as a neurodevelopmental-behavioral disorder, mainly diagnosed in early development and characterized by some stereotypes of behavior presented in several places, such as home and school [1]. ADHD is the most prevalent neurobehavioral disorder diagnosed in youth, affecting 5–7% of school-aged children globally [2,3,4,5,6]. Recent research has highlighted an increased rate of co-morbidity between ADHD and a variety of psychiatric disorders, such as mood, anxiety, and conduct disorders. Consequently, ADHD is most likely a group of conditions, rather than a single homogeneous clinical entity, characterized by different etiologic and modifying risk factors and different outcomes [7]. There has been considerable debate about whether this overlap occurred by chance or as an artifact of referral bias, whether the comorbid conditions are secondary to the ADHD, and whether other psychiatric disorders masquerade as attentional problems. Alternatively, each of the ADHD subtypes may occur with its specific comorbidity [8]. Although a subset of ADHD children also meets the criteria for bipolar disorder (BD), the exact prevalence of this diagnosis in ADHD children is strongly debated. Regardless of its prevalence, this disorder affects a severely impaired group of ADHD children, with high rates of aggression and psychiatric disorders in their families [8].

There is a growing awareness that the onset of bipolar disorder (BD) often appears during childhood or adolescence, although the typical picture of the symptoms of adolescent mood dysregulation is in many ways dissimilar to the symptomatology of adults [9].BD is one of the most highly heritable psychiatric illnesses, second only to autism [9], with lifetime prevalence about 3% globally [10,11,12,13]. The etiology and clinical course of BD are considered to be determined by genetic and environmental factors [13]. Current models of etiology consider BD as a primarily genetic illness whose onset and course are influenced by environmental stressors, suggesting that environmental factors may either trigger or prevent the development of a psychiatric disorder [13]. The available evidence suggests that viral infections during pregnancy and adulthood, maternal smoking, climatic factors, childhood trauma, life events, and low social support might influence the onset and clinical course of BD [13]. It is worth noting that emotional abuse has the highest prevalence among adverse childhood experiences and that the impact of life events, trauma, and social support on an individual varies across countries and cultures [13]. Despite ongoing research, there is little conclusive information about the impact of psychosocial stressors on the evolution of early-onset bipolar disorder [13]. Family studies have shown a high relative risk to first-degree relatives of an individual with BD [9,14]. Twin studies further demonstrate the disorder’s heritability [9,15]. Pediatric BD is associated with various neuropsychological deficits. Attempts have been made to clarify the nature of these deficits, and to determine whether deficits are specific to mood states or whether they remain after symptomatic recovery [9]. Central findings from several small studies suggest that pediatric BD involves impairment in attention [9,16,17], set shifting [16,18], visuospatial memory [9,16,19], processing speed and interference control [17], verbal memory [17,18,20], and abstract problem solving or executive function [9,17,21,22,23]. The Performance IQ scores achieved in the Wechsler Intelligence Scale for Children (WISC)by pediatric bipolar patients were reported to be similar to those seen in schizophrenia spectrum disorders, and lower than those in ADHD and unipolar depression [9,18,22,24].

The frequent occurrence of comorbidity between the two disorders complicates further diagnosis. It is not clear whether comorbidity is explained by overlapping symptoms at the superficial level of operationally defined clinical features or whether it reflects a true overlap in disease entities at the biological level [25,26]. Although co-morbidity is often not assessed correctly because different services may proceed to clinical access differently [26,27,28], the patterns of overlap with other disorders can be informative about how clinicians conceptualize disorders [26,27,28]. BD and ADHD co-occurrence has been extensively studied by numerous studies. Some researchers concluded that comorbid ADHD/BD is familiarly distinct from other forms of ADHD and may be related to what others have termed as childhood onset BD (characterized by irritability rather than euphoria, a chronic rather than episodic course, and a severe mood dysregulation leading to marked impairment) [9,29]. Retrospective studies in adults have led to similar conclusions, suggesting that ADHD symptoms represent a prodromal or early manifestation of pediatric-onset BD in certain at-risk individuals [9,30]. A prospective study of the rate, risk, and predictors of switching from ADHD to prepubertal and early adolescent bipolar I, showed a higher risk for individuals with ADHD (the cumulative risk of switching from ADHD to bipolar I through the 6-year follow-up was 28.5%, compared to a 2% switching rate for healthy controls) [9,31].

## 2. Materials and Methods

### 2.1. Design

We present a two-year follow-up of a bipolar 15-year-old female patient with a previous diagnosis of ADHD during childhood, discussing a number of aspects of concern that emerged during diagnosis and treatment of BD. The study attempts to describe the early-onset form of the BD and to discuss the controversies occurring between the two diagnoses, BD and ADHD, in younger populations.

### 2.2. Assent and Consent

Decision-making in the framework of child and adolescent psychiatry presents a variety of challenges for children, parents/guardians, and child and adolescent psychiatrists (CAPs) concerning the ethical practice related to standards of care and the obtainment of the patient’s consent. In our clinical and research case, decision-making relies on the concepts of assent and consent by the relatives (Principle IV of the AACAP Code of Ethics, 2014) [32,33,34,35,36]. Parents/guardians and CAPs did not exclude the girl from the decision-making. They recognized that, because of her developmental level, the child might not be capable of according a completely reasoned consent; however, she might be capable of having preferences and communicating them. The importance of her involvement in the decision-making process was also considered a serious factor, recognizing that the girl’s level of participation was lower than that required [32,33,34,35,36]. The patient was given an Informed Assent Form and the parents were given an Informed Consent Form. They both provided their written consent to the content of the documents, after an explanation of the procedure. The patient’s anonymity has been protected.

### 2.3. Assessment Approach

Acknowledging the difficulty of diagnosing early-onset BD, we followed a comprehensive assessment approach [22,32,33] including: (1) a timeline of the child’s development, from birth to the present, showing all prior mood episodes;(2) a semi-structured clinical interview, including co-morbid conditions;(3) a family history genogram to ascertain familial loading;(4) depression and mania rating scales to assess symptom severity and track treatment outcome via administration of the 17-item Hamilton Depression Rating Scale (HAMD-17) [37] and Young Mania Rating Scale (YMRS) [38]; (5) global rating scales using multiple informants (parents, teachers) [34]; and(6) use of mood logs [39].

## 3. Results

### 3.1. Case Description

A15-year-old female patient with ADHD symptoms for the past six years presented worsening irritability in the last four months. Assessing the patient’s history, interesting data were revealed regarding genetic and environmental factors that might have been related to the development of the psychopathology.

The patient lives with her parents in a small provincial town and is a student in the secondary education. Both patient and parents reported that during her preschool age the family experienced extreme financial instability because both parents were unemployed for a long time, which “led them in despair” after the loss of the child’s paternal grandfather, who was the main supporter of the family. This financial insecurity led the family to change location several times before the girl reached the age of five years old. Parents described this period as very difficult, with tensions, frequent frictions between themselves, and outbursts of anger directed towards the child. The patient recalled traumatic events of her childhood, such as physical and emotional abuse by her parents. We highlight that the trauma history mentioned by the patient was not included in the parents’ reports, although such trauma is supposed to have the highest prevalence among adverse childhood experiences. Both parents, during the assessment, underestimated her descriptions of traumatic events, characterizing them as “rarely occurred” or “insignificant to mention”, emphasizing their absence of empathy.

The girl did not consult any child psychiatry service until the age of nine to ten years old, when she was “suspected” to suffer from attention deficit hyperactivity disorder (ADHD) and treated with methylphenidate, which was discontinued because of a lack of beneficial effects. The girl had no psychiatric history. She had not previously suffered from any physical illnesses, nor used drugs, alcohol, or medication. Her family history was positive via her maternal aunt, who had been diagnosed with recurrent BD since her adolescence.

The patient was admitted to our service after presenting symptoms such as grandiose beliefs, agitation, and distractibility for the previous two days. She did not require sleep, and spent nights thinking about excessive plans, such as becoming a model. The symptomatology rendered her totally unable to take care of her basic needs, such as hygiene and school attendance. There were no recent stressors. The mental status examination revealed a child that was unable to remain seated, was excessively active during the interview, and presented logorrhea, accelerated speech, and flight of thoughts. She expressed countless projects and grandiose beliefs. She was irritable, denying hallucinations and suicidal ideations.

The patient was diagnosed with acute manic episode with no identifiable organic etiology but with a possible biological vulnerability for emotional dysregulation, given the maternal aunt’s BD. ADHD was considered part of the differential diagnosis. However, the clinical presentation—particularly the mental status examination as previously described—clearly corresponded to the severity of a mood disorder. In addition, the manic episode also heightened the prior hypothesis of her biological vulnerability to emotional dysregulation.

The patient was referred to a child psychiatric clinic for treatment. Baseline metabolic tests were negative, including complete blood count, electrolytes, thyroid levels, and urine analysis. A physical evaluation by the pediatrician yielded no significant findings. Electroencephalogram and brain magnetic resonance imaging (MRI) were negative. After no improvement with quetiapine (twice daily dosing totaling 600 mg, p.o.) and sodium valproate (1200 mg daily in divided doses, p.o.) she was also treated with olanzapine (5 mg, p.o.). Medication and the strong presence of nurses and psychoeducators stabilized her condition in fewer than ten days. Hospitalization lasted one month to monitor her symptoms and level of functioning during her progressive return to school. The patient was discharged, with a follow-up by a CAP.

During subsequent weeks, her compliance with medication prevented relapse. This coincided with the stability that occurred in her family environment. Her parents were supportive and showed empathy regarding her fragile emotional state and her preference to internalize her feelings. No pharmacological change was made and, with the presence of the multidisciplinary team, her condition remitted in the following months.

### 3.2. Diagnosis

The patient as a bipolar family member, from a family with an earlier-onset proband (aunt), was more likely than others to have an early onset [9,27]. In this case, the early onset of BD represents a diagnostic complexity because the prodromal symptoms must be evaluated in the context of an ADHD. Acknowledging the difficulty of diagnosing early-onset BD, we followed a comprehensive assessment approach [22,40,41] as mentioned above. The Diagnostic and Statistical Manual of Mental Disorders (fifth edition; DSM-V) spells out the criteria for diagnosing a manic, hypomanic, and major depressive episode. Different combinations of these episodes are required to diagnose BD I versus BD II [1].

### 3.3. Symptoms

Semi-structured interviews were used to examine the disorder’s clinical presentation and to provide a particularly helpful description of how symptoms such as euphoria, grandiosity, and hypersexuality manifested differently in childhood and adolescence. For example, a common presentation of grandiosity in childhood is to instruct teachers about how the class should be taught. In adolescence the patient expressed grandiose delusions, such as the achievement of a prominent profession while failing at school. Initially, the main symptoms were increased irritability and core symptoms of ADHD (hyperactivity, distractibility, impulsivity, and restlessness) that persisted for six years. Symptoms that were revealed later included, during the first manic episode: intense energy; distractibility and pressured speech; affective and psychotic symptoms, such as abnormally elevated mood and mixed emotional and psychotic symptoms (ideas of persecution—“I’ve being chased by a black Mercedes”); increased irritability and grandiosity; increased activity; distractibility; impulsivity; and restlessness. Other symptoms revealed later included: risky behaviors, such as substance misuse (alcohol), staying out late, defiance, and aggressivity; decreased need for sleep; and risky sexual misbehavior (masturbating in public places, such as the school hall).

Although the patient expressed no suicidal thoughts, the CAP assessed these symptoms in detail, because other studies highlighted high rates of suicidal ideation among adolescents with BD, which may reveal increased suicidal risk in this particular group of patients and suggest the need for close monitoring [17,42]. During assessment, symptoms such as depression, guilt, self-blame, hopelessness, suicidal ideation, psychomotor retardation, and other depressive-type cognitions, were not reported. Through the developmental history investigation, speech, language, and coordination issues during preschool age were mentioned in agreement with the pre-existing ADHD diagnosis.

### 3.4. ADHD and BD Comorbidity

Increased rates of symptoms commonly described as part of these syndromes have been reported in children with both ADHD and BD [7,8,22,26]. Distinguishing BD from ADHD was needed, because symptoms such as hyperactivity and distractibility are criteria for both disorders. Differentiation of BD from ADHD is critical for clinicians because the management of the two is different and also very difficult because some of the symptoms common in mania, such as hyperactivity or distractibility, also occur frequently in ADHD [43].

### 3.5. Treatment

Treatment required a multimodal approach that combined pharmacotherapy and psychotherapy [22,41], and the patient and parents’ involvement in the design of the treatment framework was crucial. Medications and adjunctive psychoeducational psychotherapy have a demonstrated efficacy in treating BD, as follows:

#### 3.5.1. Pharmacotherapy

The appropriate medication for the treatment of a BD should be based on patient’s specific factors, such as physical history, previous response (if applicable), and medications prescribed, in addition to family and patient’s preferences, because they affect the treatment’s compliance [44,45,46,47]. According to the parents’ reports, during the onset of symptoms—six years earlier—the treating physician proposed psychostimulants for ADHD to enhance cognitive capacity, attention, and concentration, although the parents preferred to follow only psychotherapy as a treatment. The combination of the poor cooperation and compliance of the child, with the fear of the parents about the impact and side effects of an early chronic administration of psychostimulants, resulted in the absence of any medication until the present examination.

##### Acute Manic Episode

When the first manic symptoms appeared, the patient was not yet treated with psychostimulants that could justify the episode as resulting from side effects. For acute manic symptoms mixed with psychosis, initial treatment was combined with a mood stabilizer and an atypical antipsychotic. The dose was increased as rapidly as possible to achieve the lowest therapeutic dose that would produce the desired clinical effect:Quetiapine (atypical antipsychotic: twice daily dosing totaling 600 mg, p.o.) appeared to be effective in combination with sodium valproate (mood stabilizer: 750–1500mg daily in divided doses, p.o.) [46,48].A good response was also observed in an open-label trial of olanzapine (atypical antipsychotic: p.o., 5 mg trials) administered orally when manic symptoms were exacerbated [46,49].

##### Relapse Prevention

During her follow-up, mood stabilizing medication halted the rapid shift from high to low moods and back again, and was particularly useful in preventing manic episodes. Antipsychotic medications also acted as mood stabilizers and synergistically decreased episodes that she was experiencing as “a break in reality—an inability to distinguish what’s real from what isn’t”. Olanzapine was used solely for acute treatment, the dosage was reduced and discontinued (tapering over 2 weeks or more) after full remission of symptoms within 3 months. Quetiapine and sodium valproate have been shown to be effective (or probably effective in relapse prevention) and are appropriately continued as long-term treatment is planned (quetiapine: twice daily dosing totaling 400–600 mg, p.o.; and sodium valproate: 750–1000 mg daily in divided doses, p.o.).

Continuous monitoring of body mass index, blood pressure, fasting glucose levels, lipids, serum drug levels, and hepatic and hematological indices took place during the treatment. Once the patient was stabilized on an appropriate dosage of medication, maintenance treatment continued with the goal of relapse prevention. Ongoing monitoring for symptom recurrence, including suicide risk, was important [46,50].

#### 3.5.2. Psychotherapy

Several psychosocial treatments for BD have been shown to be effective in adolescents. Effectiveness is generally measured as longer time to relapse, more time being well, improved functioning, and/or fewer or less severe symptoms/episodes [9]. These included:

##### Psycho Education

An approach that involved individual and family teaching to enhance understanding of the disorder (symptoms, classification, etiologies, course, and prognosis). Provided information about medication adherence (classes of medications, alternative therapies, withdrawal syndromes, risks of nonadherence). Developed approaches to detect new episodes (detection of prodromes, warning signs of relapse, relapse prevention planning) and identified illness coping strategies (stress management).

##### Cognitive-Behavioral Therapy (CBT)

A skills-based individual treatment, which helped the patient recognize and modify the connection between maladaptive thoughts and moods. Structured exercises were applied to identify (thought records, mood diaries, activity scheduling) and modify maladaptive thoughts and behaviors. Focused on automatic negative thoughts, distorted thinking, and maladaptive schema [51].

##### Family-Focused Therapy

An approach that involved both the patient and her family and consisted of psychoeducation about BD, and training in communication and problem-solving skills, strengthening the empathic parental attitude [52].

### 3.6. Outcomes

A two-year interdisciplinary follow-up and therapeutic intervention gave us the opportunity to assess the comorbidity of the two pathologies. Our patient recovered (defined as eight consecutive weeks with minimal to no symptoms) from her index episode and had no recurrence in the following two years.

## 4. Discussion

### 4.1. Aspects of Concern

In spite of the precision with which the DSM lays out criteria for diagnosing BD, the clinical presentation of the disorder in children and adolescents is widely debated. Areas of controversy include whether the diagnosis of BD in youth should require clearly demarcated mood episodes and, if so, of what duration, and whether specific hallmark symptoms (euphoria and grandiosity) should be required. The National Institute of Mental Health Research Round table on prepubertal BD (2001) agreed that pediatric BD can be described with “broad” or “narrow” phenotypes [9]. The *narrow phenotype* is characterized by recurrent periods of major depression and mania or hypomania fitting the classic definitions of BD I and II, respectively [9]. The *broad phenotype* has been variously defined but may involve chronic mood lability/instability rather than discrete mood episodes, and irritability with no euphoria or depression [9]. Children with the broad phenotype constitute the majority of referrals to clinicians and are characterized by severe irritability, “affective storms,” mood lability, severe temper outbursts, depression, anxiety, hyperactivity, poor concentration, and impulsivity, all with or without clear mood episodes. It is unclear whether broad phenotypes among children are true precursors to full bipolar I disorder in adulthood [9,53].

In our case, comorbidity of BD with ADHD was interesting to study because we had to assess the following details:ADHD onset diagnosed under the age of 13 years old, and thus likely to be pre-pubertal, which is in keeping with reports of an earlier onset of BD in youth also suffering from ADHD [17,54]. In contrast, high rates of ADHD are present in children with BD in US studies, with overlap rates from 57% in adolescents to 98% in children [17,54,55]. The reported rates of BD in children with ADHD are higher with an onset in childhood than in adolescence [17,25].The patient had both conditions even after the remission of overlapping symptoms and continued to meet diagnostic criteria for both conditions, and the predominant mood of mania was irritability rather than euphoria [17,29,56,57].No developmental and global learning difficulties were detected. Conversely, children with ADHD have been found to have more specific developmental and global learning disabilities compared to those with BD. The latter finding is consistent with the well-established literature about the patterns of comorbidity among children with ADHD [58] and was useful in distinguishing between ADHD and BD [17].In our case study, symptoms such as hyperactivity, distractibility, impulsivity, and restlessness were related to ADHD, as would be expected given that these are core symptoms of the disorder. Additionally, abnormally elevated mood, decreased need for sleep, and psychosis were related to BD. In the literature it is reported that elation and decreased need for sleep may be good symptoms for distinguishing children suffering from BD from those suffering from ADHD [9,17,59,60]. In our case study, irritability was not useful in the differential diagnosis of BD and ADHD. Irritability as a criterion for BD must be distinguished from irritability as a normal developmental phenomenon and as a common non-specific psychiatric symptom [9]. Indeed, the nosologic status of irritability in children and adolescents is at the center of the debate about the phenomenology of BD in youth [16,61,62,63]. Many researchers suggest particular caution is required concerning the use of irritability as a criterion for BD in children and adolescents [17,64,65,66].

In this clinical case, several aspects of concern about the diagnosis, treatment, and prognosis of BD emerged. Diagnosis of pediatric BD has serious implications for significant short- and long-term morbidities in young people, including poor academic and social performance, psychosocial dysfunction, and increased risk of suicidal behavior [7,42,67,68]. Accurate differential diagnosis of childhood BD from ADHD is crucial for the appropriate choice of medication and psychosocial interventions [17]. Moreover, a diagnosis of BD may also lead to treatment with antipsychotics and/or mood stabilizers; both medications may be associated with greater risk of adverse effects in youths [17,69]. Although these treatments have been used increasingly in children, including preschoolers [70], the risk of excessive pharmacotherapy cannot be understated [17]. Another aspect of concern is the impact of diagnostic label on a child’s self-conception and emotional life [17]. Although pharmacotherapy is the mainstay of treatment for BD, medication offers only partial relief for patients. Exclusive treatment with pharmacologic interventions is associated with disappointingly low rates of remission, high rates of recurrence, residual symptoms, and psychosocial impairment. Bipolar-specific therapy is increasingly recommended as an essential component of illness management [70]. Psychotherapy hastens the recovery from depressive episodes, prevents new mood episodes, and also contributes to the improvement of functioning and quality of life. Given the relatively modest risks associated with psychotherapy (i.e., loss of confidentiality) and robust benefits, psychosocial treatments are considered an important component of BD illness management.

Regarding aspects of concern about the prognosis of BD, long-term follow-up studies illustrate the low recovery rates and high relapse rates associated with the illness. Juveniles have a slower return to euthymia, but a lower relative risk of relapse, and longer time in remission than adult BD patients [9]. Childhood onset is a predictor less associated with recovery than adolescent onset [9]. Many studies point out dysfunctions associated with early-onset BD, with negative outcomes including high rates of hospitalization; suicidal behavior; psychosis; reckless behavior; aggression; substance abuse; utilization of psychiatric, medical, and educational services; severe family conflicts; significant caregiver burden; and chronic psychosocial impairment [9,71,72,73,74,75]. Some studies make a case for continuity between child- and adult-onset BD based on the similarity of mania symptom distribution between the two, the occurrence of both within the same families, the occurrence of maternal warmth and psychosis as predictors of both outcomes, and the fact that, across the life span and especially in youth, BD usually follows a changeable and sinuous course [9,26,76]. Factors such as baseline psychosis and low maternal warmth predict earlier relapse to mania or hypomania [9,76].

### 4.2. Merits and Limitations

The major contributions of this case report are its emphasis on the narrative aspect (in-depth understanding) and its educational value in helping raise pertinent questions regarding phenomenology, onset, co-morbidity, duration, and management of this important group of disorders [77]. On the contrary, the major limitations of this case report are the lack generalizability of the results, inability to establish a cause–effect relationship, and the danger of over-interpretation or/and misinterpretation and generalization without justification. In addition, publication bias could be a limiting factor because researchers (or journals in general) favor positive-outcome findings [77,78,79]. Retrospective design may be a methodological limitation because case reports are written after the relevant event, i.e., the observation. Thus, retrospectively, the medical record might not contain all of the relevant data and recall bias might prevent us from obtaining the necessary information from the patient, family members, and health professionals [77].

The clinician (first author) determined that the case report method was the appropriate article type. It is hoped this report can act as a stimulus in the continuing debate regarding the medical case reporting of ADHD and BD comorbidity. The case presentation was conducted by the first author and was cross-checked at every stage by the second and fourth authors to ensure accuracy and validity of data and results, respectively.

## 5. Conclusions

Despite the occurrence of major risk factors, such as early onset and positive family history, the patient’s condition improved rapidly without any relapse in the subsequent two years. Early and differential diagnoses are considered crucial protective factors for ADHD, leading to an appropriate planning of treatment. In addition, parental involvement and empathic attitude towards the patient supported the latter to cooperate and comply with the treatment, enhancing positive outcomes and stability. Research is required into the reliability and validity of diagnostic protocols and criteria for BD in children and adolescents, and also into the development of individualized therapeutic planning.

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
