# Peer review of "Childhood ADHD and Early-Onset Bipolar Disorder Comorbidity: A Case Report"

_brainsci, 2020, doi:10.3390/brainsci10110883_

Round 1
Reviewer 1 Report
Paraskevi et al. presented a case report of a two-year follow-up of a 15-year-old female patient who was diagnosed with bipolar disorders (BD) and previously diagnosed with attention-deficit/hyperactivity disorder (ADHD) during childhood. The authors discussed the diagnosis of comorbidity and its treatment. The manuscript could be acceptable upon major revisions.
Major comments.
- The objectives are unclear. It is not explicitly stated in the manuscript the aims of this study. From the title, it is supposed to discuss the risk factors of BD and ADHD comorbidity. However, from the introduction, it seems to target the consequences of the comorbidity and its impact on the physical and mental development of adolescents. Surprisingly, in the methods, it seems the author would like to dive into the complexity of diagnostic criteria and the efficacy of treatment.
- The novelty of this study is limited. Please clarify how the case report may add value to the literature.
- Please describe the ethical practice as per the standard of care and the obtainment of patient consent.
- Case description. Many case descriptions are missing, e.g. appropriate details of the case such as socioeconomic characteristics, and mentions interventions in detail such as doses, timing, route of drugs.
- Discussion. Please add an adequate literature review pertinent to the case. It is also important to mention the limitations related to the case.
- Conclusion. Add implications of a case with a core key message.
Minor comments.
- Please revise the title to reflect the nature of the study as a case report.
Reviewer 2 Report
First of all, congratulations to the authors for addressing such an important and complex topic from a clinical perspective.
Overall, the article is well written and addresses the main issues related to the difficulty of diagnosing ADHD and early-onset BD. Anyway, I have some doubts and suggestions that I think should be better clarified in the paper:
1-The introduction makes reference to important points related to the two diseases, however the importance of environmental factors in the development of BD is missing, focusing mainly on the genetic load. Similarly, the presence / absence of these risk factors is not described in the clinical case.
2-In the Materials & Methods section, the DESIGN subtopic should be revised. In that same topic there is reference to the terms "soft signs" of BD that are not defined throughout the paper.
3-The description of the clinical case should be included in the results section and not in the "MATERIALS & METHODS" section. Likewise, the clinical description of the case must be more detailed, namely with regard to the order of appearance and evolution of each symptom. The treatment needs to be better clarified, especially with regard to pharmacotherapy. It remains to be clarified whether, when manic symptoms started, the patient was treated with psychostimulants that could justify the manic episode as secondary to this therapy.
Round 2
Reviewer 1 Report
The authors have addressed all my comments.